# Exploring the Use of Mobile Health for the Rehabilitation of Long COVID Patients: A Scoping Review

**DOI:** 10.3390/healthcare12040451

**Published:** 2024-02-10

**Authors:** Kim Daniels, Joanna Mourad, Bruno Bonnechère

**Affiliations:** 1Department of PXL-Healthcare, PXL University of Applied Sciences and Arts, 3500 Hasselt, Belgium; kim.daniels@pxl.be; 2REVAL Rehabilitation Research Center, Faculty of Rehabilitation Sciences, Hasselt University, 3590 Diepenbeek, Belgium; joanna.mourad@uhasselt.be; 3Technology-Supported and Data-Driven Rehabilitation, Data Sciences Institute, Hasselt University, 3590 Diepenbeek, Belgium

**Keywords:** telerehabilitation, COVID-19, digital healthcare, post-COVID care, health-related consequence, rehabilitation, COVID-19 sequelae

## Abstract

The COVID-19 pandemic has led to a substantial revolution in the incorporation of digital solutions in healthcare. This systematic review investigates the enduring physical and psychological consequences individuals experience up to two years post-recovery. Additionally, it focuses on examining the influence of mHealth interventions on these effects. Significantly, 41.7% of survivors experience lingering symptoms that have not been addressed, while 14.1% encounter difficulties in returning to work. The presence of anxiety, compromised respiratory functioning, and persistent symptoms highlight the immediate requirement for specific therapies. Telehealth, particularly telerehabilitation, presents itself as a possible way to address these difficulties. The study thoroughly examines 10 studies encompassing 749 COVID-19 patients, investigating the efficacy of telerehabilitation therapies in addressing various health markers. Telerehabilitation-based breathing exercises yield substantial enhancements in functional performance, dyspnea, and overall well-being. The results emphasize the potential of telerehabilitation to have a favorable effect on patient outcomes; however, more research is needed to strengthen the existing evidence base, as one of the most important limitations is the limited number of trials and the evaluation of varied therapies. This analysis highlights the significance of digital solutions in post-COVID care and calls for ongoing research to improve the comprehension and implementation of telehealth interventions in a swiftly changing healthcare environment.

## 1. Introduction

COVID-19 as defined by the World Health Organization (WHO) is an infectious disease caused by the SARS-CoV-2 virus, significantly impacting the healthcare system, including rehabilitative care [1]. As the world navigates the multifaceted challenges posed by the COVID-19 pandemic, the repercussions extend beyond the acute phase, giving rise to a critical need for sustained healthcare strategies.

It has been indeed shown that while the majority of individuals affected by COVID-19 experience mild-to-moderate illness, around 10% to 15% develop severe symptoms, with 5% reaching a critical state [2]. The duration of recovery from COVID-19 varies depending on the severity of symptoms, averaging from two to three weeks [3]. While pulmonary impairments represent the most common manifestation of COVID-19, there is a plethora of symptoms originating outside the respiratory system (e.g., cardiovascular, hematologic, renal, central nervous system, gastrointestinal, and psychosocial manifestations) [4]. There is a growing body of evidence supporting the notion of extra-respiratory dissemination of coronaviruses. Although the majority of individuals recover fully within 12 weeks post-infection, a significant proportion of patients with history of COVID-19 may encounter various long-term health consequences.

In addition, the pandemic has brought unprecedented challenges to global health and economies, catalyzing a transformative shift toward digital solutions across various industries [5]. Notably, education swiftly embraced remote learning strategies, and health care organizations rapidly adopted digital solutions and advanced technologies to overcome pandemic-induced challenges [6]. While digital tools initially addressed acute needs, the ongoing evolution of these solutions presents an opportunity to define and adopt new models of care. In fact, as a result, the long-anticipated digital transformation in healthcare was accelerated, a change that was previously slower to materialize despite constant introductions of new technologies. The list of emerging digital solutions is expanding rapidly in health care, encompassing video visits, mobile apps, wearable devices, chatbots, artificial intelligence-powered diagnostics, voice-interface systems, and mobile sensors. This digital revolution has given rise to a new category of services, including the oversight of individuals in home quarantine and large-scale population surveillance.

The International Classification of Functioning (ICF) [7] is of utmost importance in the ever-changing digital healthcare environment and the urgent requirement for continuous healthcare initiatives in the aftermath of the COVID-19 pandemic. The ICF offers a complete structure for comprehending health and states associated to health [8].

The imperative to combat COVID-19 and the anticipation of recurring waves have underscored the need to review and sustain the digital technologies implemented during the crisis period.

Rehabilitative care faced challenges during the pandemic due to obligated limited contacts between patients and clinicians, necessitating alternatives to face-to-face interactions. Telemedicine and remote consultation have proven effective in maintaining patient care during times when in-person interactions are restricted [9,10,11]. In fact, telehealth, or telemedicine, emerged as a viable solution, with telerehabilitation specifically addressing therapeutic rehabilitation at a distance using telecommunication technologies [12]. This innovative approach involves active collaboration between healthcare providers and patients, enabling remote prescription and the oversight of treatment while patients actively engage in rehabilitation within their daily living context.

Recent efforts sought to unravel this enigma through a comprehensive meta-analysis of survivors’ health-related consequences, shedding light on the landscape of sequelae two years post-SARS-CoV-2 infection; the most important symptoms are fatigue, sleep difficulties, dyspnea and anxiety [13]. In this meta-analysis, encompassing twelve studies with 1,289,044 participants from 11 countries, the investigation aimed to elucidate the health outcomes and sequelae experienced by COVID-19 survivors at the two-year mark. Alarmingly, 41.7% of survivors continued to grapple with at least one unresolved symptom, while 14.1% found themselves unable to resume work. The spectrum of persistent symptoms included fatigue (27.4%), sleep difficulties (25.1%), impaired diffusion capacity for carbon monoxide (24.6%), hair loss (10.2%), and dyspnea (10.1%). Furthermore, individuals with a history of severe infection exhibited a higher propensity for anxiety and experienced pronounced impairments in respiratory functions. Anxiety was notably prevalent (OR = 1.69), while impairments in forced vital capacity (OR = 9.70), total lung capacity (OR = 3.51), and residual volume (OR = 3.35) persisted after recovery. Factors contributing to a higher risk of long-term sequelae included older age, predominantly female gender, pre-existing medical comorbidities, severe infection status, corticosteroid therapy, and heightened inflammation during the acute phase. These findings underscore the substantial and lingering impact of long-term COVID-19, emphasizing the urgent need for intervention strategies. Two years after recovery, a significant proportion of survivors, comprising 41.7%, grapple with neurological, physical, and psychological sequelae. We created an ICF model summarizing the different complications induced by COVID-19 infection in Figure 1.

The imperative to preclude persistent or emerging long-term sequelae is evident, prompting a call for targeted interventions to mitigate the risks associated with long COVID-19. In the evolving landscape of digital healthcare, addressing the extended consequences of COVID-19 becomes integral to ensuring comprehensive and sustained care for affected individuals [14].

Telemedicine services can be synchronous, involving real-time discussions between patients and healthcare providers, or asynchronous, utilizing a “store-and-forward” technique for the collection and forwarding of medical history and reports; the definition of the different types of services are presented in Figure 2.

Previous reviews demonstrated the feasibility and effectiveness of telerehabilitation during the pandemic, emphasizing its ability to provide ongoing care for patients with various conditions [18]. However, the need for further research, especially in the context of post-pandemic care, was acknowledged. Other reviews specifically investigated the efficacy of telerehabilitation for the management of COVID-19 patients [19,20]. Results revealed that breathing exercises administered via telerehabilitation exhibited a significant improvement in functional scores and breathing function. Despite the low certainty of evidence, adverse events were predominantly mild or moderate, occurring at comparable frequencies in both the telerehabilitation and control groups. The findings collectively suggest that telerehabilitation holds promise in enhancing functional capacity, mitigating dyspnea, improving performance, and addressing physical aspects of quality of life, all while maintaining a low incidence of adverse events [21].

This review aims to contribute additional evidence to reinforce the findings of previous research, extending the focus to the post-COVID era. This work explores perspectives in telerehabilitation research, suggesting its potential as an added value in conventional care beyond the immediate challenges posed by the pandemic.

## 2. Methods

### 2.1. Search Strategy and Sources of Evidence

For this scoping review, several databases were consulted for studies published up to the first of December 2023. PubMed, Embase and Scopus were searched. MESH terms and title/abstract-based search terms for mobile health (‘mHealth’ OR ‘mobile health’ OR ‘applications’), COVID-19 (‘covid*’ OR ‘SARS-CoV-2′) and rehabilitation (‘rehabilitation’ OR ‘revalidation’ OR ‘reeducation’) were used to retrieve studies. References from selected papers and from other relevant articles were screened for potential additional studies in accordance with the snowball principle. The search was limited to journal articles published in English.

The PRISMA flowchart for scoping reviews was followed for screening [22]. All results were uploaded on Rayyan to filter out duplicates and then screening was performed independently by 2 reviewers based on title/abstract. Subsequently, the remaining studies were checked for inclusion based on the PICOS criteria in Table 1. In case of conflict, the two reviewers deliberated with each other.

### 2.2. Eligibility Criteria

The following PICOS criteria were defined to guide our search strategy [23].

### 2.3. Critical Appraisal of Individual Sources of Evidence

The PEDro scale, recognized as a valid and reliable tool for evaluating randomized controlled trials (RCTs), was employed for the assessment of methodological quality [24,25]. The quality of RCTs was independently evaluated by two reviewers in a blinded fashion to mitigate potential methodological bias [26]. Final decisions regarding the quality of each RCT were reached through consensus. In instances of disagreement, a third author was consulted. Consequently, RCTs were categorized into three groups: low quality (0–3/10), moderate quality (4–6/10), and high quality (7–10/10) [26].

### 2.4. Data Charting

The following information was extracted from the included studies: characteristics of the patients (age, gender, COVID status), intervention (type, number of sessions, length, frequency, duration), type of control, main outcomes, and conclusions (including feasibility and effectiveness).

### 2.5. Ethical Approval

This review was reported following the Preferred Reporting Items for Systematic Reviews and Meta-Analyses for Scoping Review (PRISMA-ScR) recommendations [22]. For the present study, no ethics committee approval was necessary.

## 3. Results

A comprehensive search utilizing various combinations of search terms yielded a total of 1678 results. Following a curation process involving the examination of titles and abstracts, 225 articles were included. The primary reasons for exclusion were the articles being systematic reviews, feasibility studies, or not directly related to the intended topic.

The remaining articles underwent a thorough full-text review, during which 215 were found to not meet the inclusion criteria. Consequently, the scoping review now comprises a total of 10 articles that met the established criteria. The complete flow chart of study inclusion is presented in Figure 3.

### 3.1. Characteristics of the Included Studies and Patients

Table 2 provides a comprehensive overview of the 10 included studies, detailing their respective characteristics. In total, 749 COVID-19 patients were included in this review: 384 in the intervention group and 362 in the control group.

The studies originated from diverse geographical locations, including China (*n =* 3), Spain (*n =* 3), the USA (*n =* 2), Brazil (*n =* 1), and Iran (*n =* 1). The majority of the studies concentrated on the target group of individuals experiencing persistent symptoms such as dyspnea after the acute phase (*n =* 8). The other included studies focused on hospitalized COVID-19 patients (*n =* 2). In total, the ten studies comprised the results of 749 COVID-19 patients; 384 of them were included in the intervention group and the other 362 in the control group.

### 3.2. Qualitity Assessment

The PEDro Scale yielded an average score of eight, indicative of a commendable level of methodological quality. The global level of evidence for randomized controlled trials (RCTs) was deemed satisfactory. A summary of the quality assessement is presented in Figure 4.

### 3.3. Evaluation Variables

Among the included studies, the evaluated aspects were divided into the classification of the ICF model: body and structure, activities, and participation.

All studies focus on the domain of body and structure. Only 30% focus on activities, and 10% focus on participation. Only one individual study focuses on the three domains of the ICF model (see Figure 5).

In the domain of body and structure, the recurrent measurements encompassed the six-minute walking test, strength assessment (e.g., hand grip), pulmonary function, dyspnea, anxiety, sleep quality, depression, cardiopulmonary fitness, symptom burden, balance, and fear. Within the activity domain, measurements included physical activity level, step count, functional evaluation, level of independence, quality of life, sit-to-stand test, and affective status. Lastly, in the participation domain, measurements such as community social support, social support, and sociodemographic information were collected. Some studies also incorporated non-specific measurements outside the ICF model, such as assessing the feasibility, usability, and satisfaction of the mHealth tools being utilized.

### 3.4. Telerehabilitation Methods

The predominant telerehabilitation approaches employed in the selected studies encompassed primarily holistic telerehabilitation applications (*n =* 4). These applications offered a comprehensive suite of features, including exercises, pulmonary rehabilitation, communication, and information. Additionally, there were applications specifically designed for pulmonary exercise alone (*n =* 3), as well as remote monitoring apps (*n =* 2) and remote guidance apps (*n =* 1). It is important to note that different terms were used in the literature; therefore, we summarized the different eHealth disciplines in Figure 5 to ease the interpretation of the results and comparison with the literature.

### 3.5. Clinical Efficacy

The comprehensive analysis of various mHealth interventions for COVID-19 recovery unveiled a spectrum of outcomes. Among these studies was the examination of applications, revealing that the significant usage of such mHealth apps correlated with enhanced physical and cognitive function, mental health, and quality of life.

The wide range of interventions and their beneficial effects on different outcomes in varied groups and situations highlight the broad range of potential applications.

#### 3.5.1. Physical Function and Exercise Capacity

Interventions focused on physical function and exercise capacity are crucial in recovering patients’ overall well-being. Churchill et al. (2023) and Rodriguez-Blanco et al. (2021) demonstrate the efficacy of structured physical therapy sessions, resulting in higher step counts and enhanced performance in functional tests [28,34]. These findings emphasize the need to start rehabilitation early and customize it to improve mobility and reduce the functional decrease commonly observed during COVID-19 recovery. Capin et al. (2022) have taken a complete strategy that highlights the potential advantages of telerehabilitation [31]. This approach provides a practical way to achieve long-term improvements in exercise capacity and functional results.

#### 3.5.2. Mental Health and Quality of Life

It is crucial to prioritize the mental health components of post-COVID-19 recuperation, as individuals frequently experience psychological discomfort after being unwell. The research conducted by Wei et al. (2020) and Liu et al. (2021) provide insights into therapies targeted at mitigating depression, anxiety, and insomnia [27,29]. The self-help tactics utilized by Wei et al. and the cognitive–behavioral therapy (cCBT) implemented by Liu et al. provide useful insights into the efficacy of focused mental health therapies. From a clinical perspective, it is crucial to include these methods in the post-COVID-19 care process to promote a comprehensive strategy that considers both the physical and psychological recovery of patients. This will ensure a patient-centered healthcare approach that addresses all aspects of their well-being.

#### 3.5.3. Sleep Quality and Fatigue

The clinical worry regarding the influence of COVID-19 on sleep quality and exhaustion persists beyond the acute period of the illness. Hajibashi et al. (2023) tackle these concerns by employing pulmonary telerehabilitation, which leads to notable enhancements in sleep quality and decreases in fatigue and anxiety [35]. For clinicians in charge of post-COVID-19 care, it is essential to acknowledge and tackle sleep disruptions and exhaustion due to their capacity to hinder recovery and contribute to lasting disability [29].

#### 3.5.4. Cognitive Function

The significance of cognitive function in post-COVID-19 treatment is typically disregarded, although it is highlighted as a crucial factor in the studies conducted by Samper-Pardo et al. (2023) and Campos et al. (2023) [33,36]. These research studies emphasize the favorable correlation between the utilization of applications, remote monitoring, and enhanced cognitive results. For therapists, it is crucial to acknowledge the possible cognitive consequences of COVID-19. It is vital to incorporate cognitive evaluations and specific therapies into rehabilitation. Recognizing the relationship between physical and cognitive health is essential for developing comprehensive care methods that effectively address the complex issues patients encounter during the recovery phase. Healthcare professionals can enhance the overall quality of life and optimize the cognitive well-being of post-COVID-19 patients by implementing therapies that focus on cognition.

#### 3.5.5. Remote Monitoring

Only one study was found about continuous remote monitoring [36]. This study specifically examined persons with chronic symptoms of COVID-19 and compared the effectiveness of remote monitoring combined with health counsel to traditional in-person therapy. Both groups exhibited enhanced fatigue and exercise capability. The remote monitoring group exhibited improvements in dyspnea, anxiety, concentration, and short-term memory.

The use of remote monitoring and health assistance, specifically for persons encountering persistent COVID-19 symptoms, demonstrated efficacy in addressing many dimensions of their well-being. In addition to the physical advantages of decreased tiredness and enhanced ability to engage in physical activity, the intervention showed favorable impacts on psychological factors, such as anxiety, as well as cognitive abilities including attention and short-term memory. This underscores the possibility of technology-driven methods in offering extensive assistance to those coping with persisting symptoms following COVID-19. Supporting the additional investigation of these discoveries could aid in enhancing and perfecting remote monitoring approaches for the ongoing management of symptoms over an extended period of time.

## 4. Discussion

### 4.1. Main Findings

Our results indicated that mHealth can be used in both patients with acute COVID-19 (i.e., hospitalized patients) and in the chronic phase (i.e., telerehabilitation). Furthermore, the combination of conventional therapy and mHealth-supported care emerged as notably more effective in improving sleep quality [29,33,35], alleviating anxiety [27,29,33,35,36], reducing dyspnea [27,29,30,32,36], and reducing fatigue/improving physical function [27,28,29,30,31,32,33,34,35,36]. The results of the different included studies provide robust support for telerehabilitation’s efficacy in addressing dyspnea, muscle strength, ambulation, and depression. While several studies accentuated the multifaceted potential of telerehabilitation in COVID-19 recovery, a contrasting viewpoint emerged, advocating for the indispensability of traditional rehabilitation interventions. Interestingly, such a type of technology can also been used to monitor the evolution of patients [37], as remote monitoring exhibited promise by fostering increased step counts, suggesting a plausible surrogate for physiological recovery [34]. The variety of approaches highlights the importance of a multidimensional strategy for comprehensive post-COVID-19 care.

These results are in line with the results of a previous review published in 2022 [20]. The findings of this study highlight several positive outcomes associated with telerehabilitation interventions across various health indicators. Breathing exercises delivered through telerehabilitation demonstrated significant improvements in the participation and function items of the ICF: 6 min walk distance, 30 s sit-to-stand test performance, Multidimensional Dyspnoea-12 questionnaire scores, and perceived effort on the 0-to-10 Borg scale, albeit with low certainty of evidence. These results underscore the potential benefits of telerehabilitation in positively impacting functional performance, dyspnea, and overall well-being, despite the need for further research to strengthen the evidence base.

In this review, we focused on COVID-19 patients, but such a kind of app can be used with patients suffering from other (chronic) respiratory conditions such as COPD (chronic obstructive pulmonary disease) [38], asthma [39], lung cancer [40], and cystic fibrosis [41]. Moreover, such a type of app can also be used with healthy subjects. For example, in a study evaluating interventions involving the Fear of COVID-19 Scale, HAMA, Pittsburgh Sleep Quality Index (PSQI), and EQ-5D-3L, significant improvements were observed in healthy individuals receiving the intervention. The intervention included an information session about COVID-19 at the study’s outset via a mobile phone video application, coupled with a breathing and relaxation exercise program conducted twice daily [42].

Another study found that breathing exercises are feasible even in healthy individuals. App users positioned their phones adjacent but not proximal to their mouths and the inclusion of onscreen text or video instructional materials did not significantly affect this distance. Participants expressed a clear preference for the app (100%, *n* = 24), citing motivation for sustained inspirations. Notably, app features like gamification (75%, *n* = 18) and breath counter (83.3%, *n* = 20) were well-regarded, emphasizing user satisfaction and engagement [43].

### 4.2. Limitation of the Scoping Review

The findings of this scoping review must be interpreted in the context of certain limitations. First, the restricted number of incorporated studies and the broad variety of treatments (such as procedures, applications, outcomes, etc.) introduce complexity to the process of comparing and extrapolating the results. We also noticed that the impact of sex and gender was almost never mentioned in the publication, even though this could have an impact on the acceptability and efficacy of the intervention. These aspects have previously been identified as facilitators or barriers to the adoption of digital technology in health [44].

While the possibility of expanding the scope of this review to include studies on alternative telerehabilitation systems or those integrating mHealth with additional devices or sensors was taken into account, doing so would have meant sacrificing the practical aspects that are inherent to mHealth, namely, its portability, affordability, and user-friendliness.

This review was limited to RCTs, but interesting results can also be found in observational studies. For example, a study analyzing 58 patients discharged from ICUs found that an mHealth program can improve health-related quality of life breathlessness, symptom burden, weekly mental health status, and COPD Assessment Test performance, indicating a positive impact on symptom burden. Additionally, there were favorable changes in mental health status, as evidenced by a decrease in the Hospital Anxiety and Depression Scale [45].

### 4.3. Future and Related Works

The research presented a wide array of mHealth therapies that have shown positive effects on different outcomes, highlighting the promise of technology in tackling the intricate issues presented by COVID-19. Nevertheless, the precise mechanisms through which these interventions exert their effects in various populations and environments are not fully comprehended. Additional studies are crucial to uncover the mechanisms that contribute to the beneficial outcomes shown in mHealth for COVID-19 recovery, as this field is still developing. Further research is required to understand the complex relationship between interventions and individual factors, in order to gain a deeper understanding that can guide the improvement and optimization of mHealth techniques for varied patients impacted by (long) COVID-19.

In this paper, we focus on the use of mHealth, while other new approaches and technology can be used to provide (tele)rehabilitation services to patients. We have seen in the methods that mHealth is part of telerehabilitation, and more evidence and publications are available concerning telerehabilitation.

Combined with external sensors: 88 individuals experiencing persistent fatigue and dyspnea post-COVID-19 underwent an 8-week supervised home-based respiratory muscle training program. Divided into inspiratory and expiratory training groups, participants engaged in 40 min daily sessions using a threshold pressure device supervised through a virtual platform. Quality of life improved significantly but not exercise tolerance. Both inspiratory and expiratory training enhanced respiratory muscle function and lower limb strength without notable effects on lung function or psychological status [46].

Other examples are serious games and virtual reality (VR). In a systematic review, authors analyze the use of VR interventions for the rehabilitation of individuals afflicted with COVID-19. The study identified a predominant emphasis on cognitive rehabilitation, with two papers (66%) employing immersion VR, while one study (33%) utilized non-immersive VR for physical rehabilitation. Notably, virtual reality was delivered to patients in the form of a game in two papers (66%). The results of this research suggest that virtual reality games have the potential to positively impact functional and cognitive outcomes, elevate patient satisfaction levels, and empower individuals to take an active role in their healthcare management. Given the unique challenges faced by COVID-19 patients, the evolving landscape of healthcare delivery, and the critical role of rehabilitation during quarantine, the exploration of novel techniques is imperative to ensure continued treatment, facilitate a return to normalcy, and ultimately improve the overall quality of life for these individuals [47].

Here, we focus on patients with COVID-19, however it is crucial to acknowledge the significant burden on the shoulders of healthcare professionals amid this pandemic, which undeniably affects their mental health. mHealth can be used to improve mental wellbeing of the clinicians. In a pilot randomized controlled crossover trial involving 34 clinicians, mHealth demonstrated above average feasibility and acceptability. Significant improvements were observed in anxiety, resilience, and patient-related burnout in the intervention group compared to the waitlisted group after one month, indicating the potential of mHealth in enhancing the well-being of healthcare professionals [48].

More generally speaking, telemedicine also exhibits promising outcomes. It has demonstrated notable improvements in alleviating psychological stress, addressing mental disorders, and significantly reducing overall stress levels among COVID-19 patients. Additionally, telemedicine was proven effective in providing benefits such as the tailored monitoring of vital parameters for home-isolated patients and facilitating clinicians in the early identification of clinical deterioration [49].

Utilizing technology and social media-based interventions emerges as a promising approach for advancing health and well-being, particularly in circumstances such as a pandemic [50]. Nevertheless, there is a recognized necessity for the formulation of guidelines regulating social media usage to mitigate potential risks and counter the dissemination of misinformation.

It is important to note that the COVID-19 pandemic accelerated the development and implementation of mHealth solutions in the field of rehabilitation. While concrete evidence regarding the management of (long) COVID patients remains limited, numerous individual studies and review are now highlighting the added values of mHealth in addressing various pathologies or conditions. This includes but is not limited to stroke, patients undergoing chemotherapy [51], patients with fibromyalgia [52], after bariatric surgery [53], hip and knee osteoarthritis [54], and urinary incontinence [55]. A consistent positive aspect found in the studies evaluating mHealth is that these interventions are well-received by patients, leading to greater adherence [56,57] and patient satisfaction [58,59].

### 4.4. Implications for the Rehabilitation

The transformative impact of the COVID-19 pandemic on societal dynamics is evident. Notably, the evolving peaks of the crisis have disrupted the assured continuity of care [60]. In response, rehabilitation services have undergone essential modifications and adaptations in their modalities of service provision and delivery to meet the exigencies imposed by the pandemic [61]. As healthcare professionals, incorporating these strategies into post-COVID-19 treatment programs could be crucial in enhancing recovery progress and reducing long-term physical impairments.

Nevertheless, several challenges still slowdown the integration of mHealth applications into clinical rehabilitation practices [62], necessitating comprehensive resolution before widespread implementation. Foremost among these challenges is the imperative need for the acknowledgment and acceptance of mHealth applications as integral components of rehabilitation interventions. The COVID-19 pandemic, while disruptive to healthcare systems, has concurrently catalyzed the development, implementation, and recognition of mHealth within clinical contexts [63]. However, it is crucial to recognize that many of the measures implemented during the crisis may be transient, and sustained efforts toward mHealth integration are contingent upon post-crisis commitment.

A pivotal obstacle lies in the current categorization of mobile solutions, which are presently subsumed under the same classifications as pharmaceuticals. This categorization not only introduces validation challenges but also poses obstacles to reimbursement processes [64]. Consequently, a paradigm shift in intervention nomenclature is imperative. Additionally, a prevalent limitation stems from the predominant development of mHealth solutions within research projects, rendering them largely inaccessible to the general patient population. This discrepancy underscores the second major impediment—the absence of social security reimbursement.

While the intricacies of healthcare system organization and participation in revalidation processes vary across countries, it is noteworthy that financial considerations and a deficiency in knowledge and experience with new technologies are universal barriers to the widespread adoption of telemedicine and telehealth. Notably, patients, irrespective of their specific pathologies or medical disciplines, consistently cite financial concerns and a lack of familiarity with technology as primary impediments [62,65]. While many patients exhibit familiarity with smartphones, apps, and mobile technology, thus mitigating concerns for a majority, this may not hold true for certain demographics, such as older adults with dementia, who may encounter substantial barriers [66,67].

In overcoming these barriers, it is imperative to not only address patient education but also focus efforts on healthcare professionals. The latter must undergo comprehensive training in the nuances of mHealth technology, including its limitations, to effectively endorse and encourage patient utilization. A valuable framework that can provide guidance in this educational endeavor is the NASSS (non-adoption, abandonment, challenges to scale-up, spread, and sustainability) framework [68]. This model examines the implementation of technology across seven domains, ranging from characteristics of the disease and patient population to funding and legislation). As the trajectory toward mHealth integration unfolds, a strategic emphasis on educational initiatives, guided by the insights provided by the NASSS framework, becomes pivotal for both patients and healthcare professionals alike.

## 5. Conclusions

The evaluated research and interventions highlight the capacity of telehealth and digital apps to have a positive influence on a range of health outcomes. Telerehabilitation therapies have shown positive trends in increasing functional performance, lowering symptom load, and promoting overall well-being in persons with chronic respiratory diseases. Significantly, the implementation of breathing exercises via telehealth resulted in notable enhancements in various clinical aspects.

Furthermore, when considering mental health and concerns connected to COVID-19, the incorporation of mobile phone applications, informational sessions, and relaxation exercises has shown significant enhancements in fear of COVID-19, levels of anxiety, and quality of sleep. The utilization of digital therapeutics has shown a level of efficacy that is comparable to conventional therapy, eliciting a pronounced inclination from users toward its distinctive attributes and gamification components.

Although these findings show promise, it is crucial to recognize the necessity for additional research to enhance the evidence and address limits in study designs and uncertainties. With the ongoing advancement of technology, adopting telehealth solutions can provide significant opportunity to improve the accessibility of healthcare, engage patients more effectively, and ultimately boost overall health outcomes. In conclusion, the data that have been examined support the further investigation and adoption of digital interventions in healthcare to enhance patient outcomes and provide comprehensive, patient-centered care.

## Figures and Tables

**Figure 1 healthcare-12-00451-f001:**
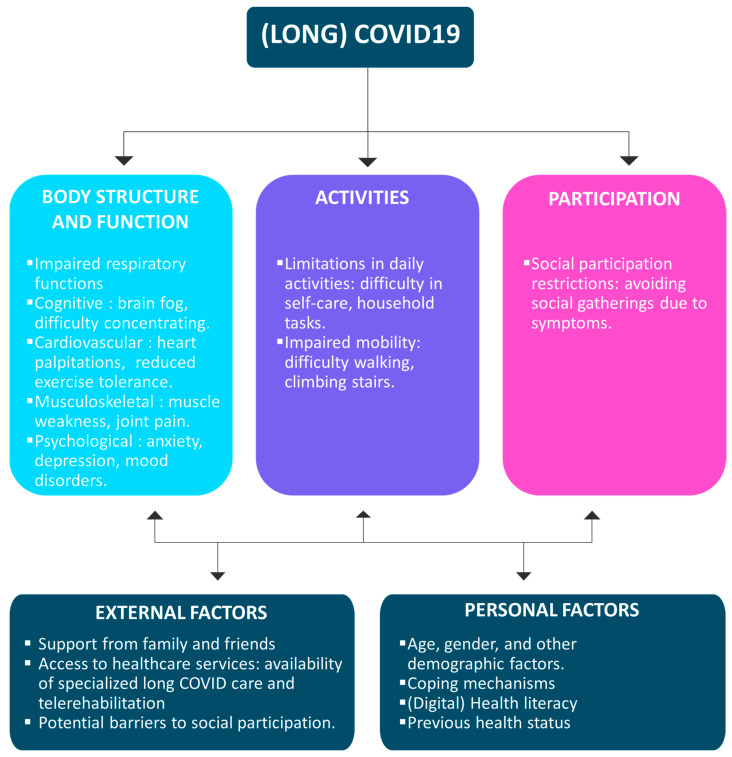
ICF model for COVID-19 repercussions and sequelae.

**Figure 2 healthcare-12-00451-f002:**
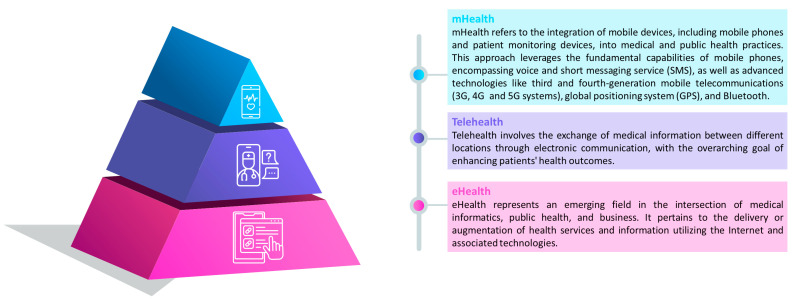
Definitions and organizations of eHealth [15], telehealth [16] and mHealth [17] services.

**Figure 3 healthcare-12-00451-f003:**
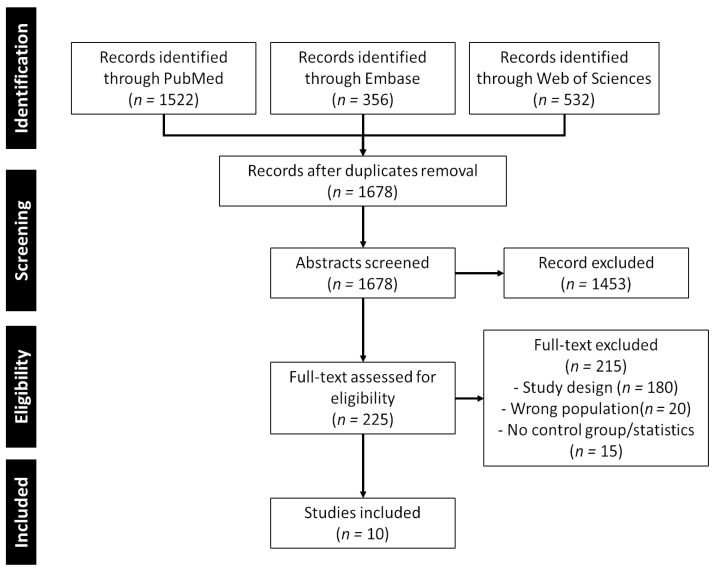
PRISMA-ScR flow chart of study selection.

**Figure 4 healthcare-12-00451-f004:**
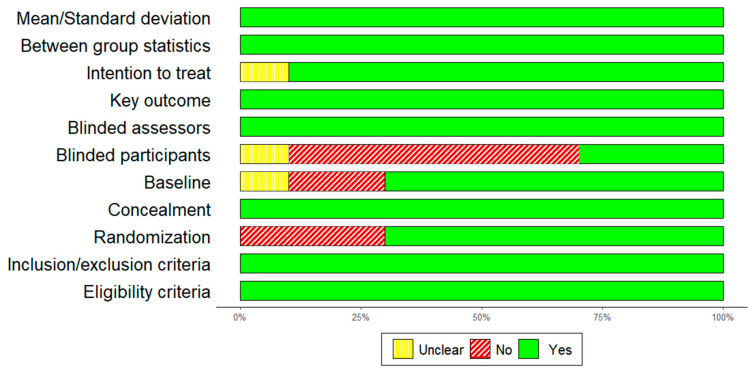
Quality of the study, author’s judgement, broken down for each item of the PEDro scale across all included studies.

**Figure 5 healthcare-12-00451-f005:**
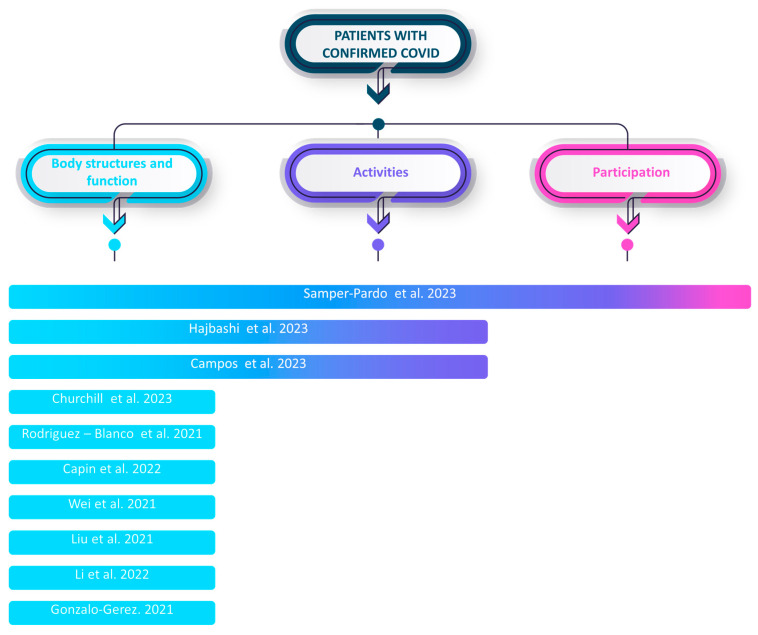
Main outcomes of the included studies according to the ICF framework [27,28,29,30,31,32,33,34,35,36].

**Table 1 healthcare-12-00451-t001:** PICOS strategy.

Population	Patients with Confirmed COVID-19
Intervention	Patient education (information about COVID-19, causes, risk factors, healthy diet, treatment modalities and exercises), or physical exercises or activities (strengthening exercises, gait training, cycling, walking, gardening, etc.) through mHealth.
Comparison	Any other intervention (i.e., physical exercises or activities, patient education without mHealth) or no intervention
Outcome	Any type of outcome measure related to the International Classification of Functioning, Disability and Health (ICF) [8]
Study Design	Randomized Controlled studies

**Table 2 healthcare-12-00451-t002:** Characteristics of the included studies.

Study, Country	Population	Intervention	Control	Outcomes	Results	Quality
**Wei et al., 2020 China [27]**	26 COVID-19 patients (40–50 years old)	*n =* 13 Self-help intervention containing four main components: breath relaxation training, mindfulness (body scan), “refuge” skills, and butterfly hug method.	*n =* 13 Supportive care	Hamilton Depression Rating Scale, Hamilton Anxiety Rating Scale	Depression and anxiety were significantly decreased in patients of the intervention group at the end of the first and second weeks when compared with the patients of the control group	9
**Rodriguez-Blanco, 2021** **Spain [28]**	36 COVID-19 patients with mild to moderate symptomatology in the acute stage	*n =* 18 Muscle conditioning telerehabilitation	*n =* 18 No physical activity	Six-minute walking test, multidimensional dyspnoea-12, thirty seconds sit-to-stand test, and Borg Scale	Both groups were comparable at baseline. Statistically significant improvement between groups (*p* < 0.05) in favor of the experimental group was obtained. Ninety percent adherence was found in our program.	6
**Liu et al., 2021** **China [29]**	252 COVID-19 patients (45–45 years old)	*n =* 126 Computerized cognitive behavioral therapy (cCBT). The system can systematically intervene in patients’ cognition, emotions, and behavior through an offline mobile terminal.	*n =* 126 Conventional treatment (periodic psychological assessments, general psychological support, and consultations discussing overall well-being and disease activity)	Hamilton Depression Rating Scale, Hamilton Anxiety Rating Scale, Self-Rating Depression Scale, Self-Rating Anxiety Scale, Athens Insomnia Scale	The cCBT group displayed a significantly decreased scores after the intervention compared to the conventional group (all *p* < 0.001). A mixed-effects repeated measures model revealed significant improvement in symptoms of depression, anxiety and insomnia during the postintervention and follow-up periods in the cCBT group	8
**Gonzalez-Gerez et al., 2021** **Spain [30]**	38 COVID-19 patients with mild to moderate symptomatology in the acute stage (18–75 years old)	*n =* 19 Pulmonary rehabilitation	*n =* 19 No physical activity	Six-Minute Walk Test, Multidimensional Dyspnoea-12, Thirty-Second Sit-To-Stand Test, and Borg Scale.	Significant differences were found for all of the outcome measures in favor of the experimental group (*p*< 0.05).	9
**Capin et al., 2022** **USA [31]**	44 Participants discharged home following hospitalisation with COVID-19 (with and without intensive care unit (ICU) stay)	*n =* 29 12 individual bio-behaviourally informed, app-facilitated, multicomponent telerehabilitation sessions with a licenced physical therapist	*n =* 15 Education on exercise and COVID-19 recovery trajectory, physical activity and vitals monitoring, and weekly check-ins with study staff.	Primary outcome was feasibility, including safety and session adherence. Secondary outcomes included preliminary efficacy outcomes including tests of function and balance; patient-reported outcome measures; a cognitive assessment; and average daily step count. The 30 s chair stand test was the main secondary (efficacy) outcome	8% (11/29) of the intervention group compared with 60% (9/15) of the control group experienced an AE (*p* = 0.21), most of which were minor, over the course of the 12-week study. 27 of 29 participants (93%; 95% CI 77% to 99%) receiving the intervention attended ≥75% of sessions. Both groups demonstrated clinically meaningful improvement in secondary outcomes with no statistically significant differences between groups.	8
**Li et al., 2022** **China [32]**	120 Formerly hospitalised COVID-19 survivors with remaining dyspnoea complaints	*n =* 59 TERECO Unsupervised home-based 6-week exercise programme comprising breathing control and thoracic expansion, aerobic exercise and LMS exercise, delivered via smartphone, and remotely monitored with heart rate telemetry.	*n =* 61 Short educational instructions	6 min walking distance (6MWD), squat time in seconds; pulmonary function assessed by spirometry; HRQOL measured (SF-12) and mMRC-dyspnea.	Adjusted between-group difference in change in 6MWD was 65.45 m (*p* < 0.001) at post-treatment and 68.62 m (*p* < 0.001) at follow-up. Treatment effects for LMS were 20.12 s (*p* < 0.001) post-treatment and 22.23 s (*p* < 0.001) at follow-up. No group differences were found for lung function except post-treatment maximum voluntary ventilation. Increase in SF-12 physical component was greater in the TERECO group with treatment effects estimated as 3.79 (*p* = 0.004) at post-treatment and 2.69 (*p* = 0.045) at follow-up.	7
**Samper-Pardo et al.** **2023** **Spain [33]**	100 Primary Health Care long COVID patients (18+)	*n =* 52 ReCOVery APP and standard therapy	*n =* 48 Treatment as usual methods established by their general practitioner	Quality of life (SF-36), Sociodemographic variables, self-reported persistent symptoms, use of ReCOVery APP, cognitive domains (MoCA), physical functioning, Affective status (HADS); Sleep quality (ISI), social support (MOS-SS); Community social support (PCSQ), Physical Activity (IPAQ-SF), personal factors	Approximately 25% of participants actively utilized the app. Results from a linear regression model indicate that increased usage time predicts enhanced physical function (b = 0.001; *p* = 0.005) and community social support (b = 0.004; *p* = 0.021). Additionally, heightened self-efficacy and health literacy are associated with improved cognitive function (b = 0.346; *p* = 0.001) and a reduction in symptoms (b = 0.226; *p* = 0.002).	8
**Churchill et al., 2023** **USA [34]**	44 participants discharged home following hospitalization with COVID-19 (with and without intensive care unit (ICU) stay) <40	*n =* 27 Physical therapy and education sessions	*n =* 14 Weekly check-in calls	Demographics, Physical function testing, a health diary via Fitbit Steps	Step counts increased in favor of the intervention group (*p* < 0.001) culminating in an average daily step count of 7658 (*p* < 0.001) at the end of week 3. During the remaining 9 weeks, weekly step counts increased by an average of 67 (*p* < 0.001) steps per week, resulting in a final estimate of 8258 (*p* < 0.001)	8
**Hajibashi et al., 2023** **Iran [35]**	52 Discharged COVID-19 patients (18–65)	*n =* 26 pulmonary telerehabilitation and progressive muscle relaxation for 6 weeks	*n =* 26 pulmonary telerehabilitation fir 6 weeks	Functional capacity and secondary (dyspnea, anxiety, depression, fatigue, sleep quality, and quality of life	The experimental group showed significantly higher sleep quality (*p* = 0.001) and significantly lower fatigue (*p* = 0.041) and anxiety (*p* = 0.001) than the comparison group. No between-group differences were observed in terms of other outcomes (*p* > 0.05)	10
**Campos et al., 2023** **Brazil [36]**	37 Adults with persistent symptoms of COVID-19	*n* = 15 Remote monitoring with health guidance	*n = 22* Face to face rehabilitation 8 weeks (2/week)	Fatigue, dyspnea, and exercise capacity, Lung function, functional status, symptoms of anxiety and depression, attention, memory, handgrip strength, and knee extensor strength were secondary outcome measures	Both groups showed improved fatigue and exercise capacity. Exercise rehabilitation improved dyspnea, anxiety, attention, and short-term memory.	8

## Data Availability

Not applicable.

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
