# Peer review of "Exploring the Use of Mobile Health for the Rehabilitation of Long COVID Patients: A Scoping Review"

_healthcare, 2024, doi:10.3390/healthcare12040451_

Round 1
Reviewer 1 Report
Comments and Suggestions for Authors
Overall, the manuscript is interesting and well-written. There are some issues which can be addressed by the authors.
1- The introduction section is a bit long. Please make it shorter.
2- Please ensure that the aim of the study has the same wording in the abstract and introduction.
3- In the methods section, please start with the study design.
4- As it is a scoping review, please use PRISMA flowchart and the subheadings recommended by PRISMA-ScR in the methods and results sections.
5- The number of the searched databases is very limited.
6- Please re-organize Table 2 chronologically.
7- In the limitation of the research, the authors noted “Limitation of the systematic review”, please change it to the scoping review.
Author Response
See attached document

Reviewer 2 Report
Comments and Suggestions for Authors
The manuscript " Leveraging Mobile Health for Rehabilitation of Long-COVID Patients: A Scoping Review" is a systematic review to understand the long-term impact of COVID-19 and the significance of mHealth interventions on long-term sequelae of COVID-19.
This study is particularly relevant in the current context, especially with the emergence of long-term sequelae of COVID-19 being a significant research focus. The exploration of the utility of mHealth, as is, or in combination with conventional therapies, is prudent and is particularly relevant in the digital health context.
The manuscript is well put together and easy to follow. The methodology is fairly robust and well-described. The figures and tables are relevant and help aid the flow of the manuscript. Figure 3 – red/green needs to be changed to be made accessible to color-impaired readers. While the study design is good, there being only 10 relevant studies does limit the power of the systematic review.
While the authors provide a table that describes the individual studies - results and outcomes and also scored the quality of RCTs in a fairly independent fashion, since there are only 10 studies, further exploration of the particularities of individual studies, such as unique findings in the context of different successful mHealth interventions would enhance the manuscript.
The authors state that mHealth interventions would benefit both acute and long-term COVID treatment and also state that when telehealth interventions are combined with traditional rehabilitation efforts, they are more effective. This is a significant finding though not unique to this review. Another significant point of the study is that clinical efficacy varies, leading to a “ spectrum of outcomes” with the different mHealth interventions.
I would encourage the authors to highlight unique points of the individual studies that have resulted in the benefits stated, particularly related to the implementation of the mHealth interventions in RCTs. The authors point out that significant usage of mHealth apps correlated with enhanced physical and cognitive function and other benefits.
An exploration of this aspect of what factors lead to “significant usage” in discussion is particularly relevant and is often ignored and, if done, would enhance the manuscript. While the main findings are robust, the discussion could benefit from a deeper exploration of the mechanisms behind the observed benefits of mHealth.
Overall, the authors have generated a useful dataset/manuscript with multifaceted health outcome data and have done a thorough analysis, with adequate care taken in the study design. The discussion and claims are largely in keeping with the results obtained. I think some additional enhancements, as described, would greatly strengthen the manuscript.
I would recommend the study for publication with some minor changes.
Comments on the Quality of English LanguageThere needs to be another review for grammatical errors. Overall, the quality of the English is excellent, with clear, coherent, and professionally articulated text that meets the high standards expected in academic publications.
Author Response
See attached document

Reviewer 3 Report
Comments and Suggestions for Authors
I am sending a detailed review in the attachment.
Kind regards,

Author Response
See attached document

Reviewer 4 Report
Comments and Suggestions for Authors
The mini-review (including 10 randomized control trials) discussed interventions (physical exercise, gait training, etc. ) and compared them across any different kinds of intervention. The review demonstrated the effectiveness of telehealth and digital applications in positively impacting various health outcomes.
The review concluded that telerehabilitation therapies have been effective in improving functional performance, reducing symptoms, and enhancing well-being in individuals with chronic respiratory illnesses.
The study is well-written and the graphics are not. I have only one minor comment:
Comment1: please discuss the impact of sex and gender on the intervention results in the article
Author Response
See attached document

Round 2
Reviewer 1 Report
Comments and Suggestions for Authors
I appreciate the authors for their time and efforts to revise the manuscript and addressing my comments.